# Using AE Signals to Investigate the Fracture Process in an Al–Ti Laminate

**DOI:** 10.3390/ma13132909

**Published:** 2020-06-29

**Authors:** Grzegorz Świt, Aleksandra Krampikowska, Tadeusz Pała, Sebastian Lipiec, Ihor Dzioba

**Affiliations:** 1Department of Strength of Materials, Concrete Structures and Bridges, Faculty of Civil Engineering and Architecture, Kielce University of Technology, Al. 1000-lecia PP 7, 25-314 Kielce, Poland; akramp@tu.kielce.pl; 2Department of Machine Design, Faculty of Mechatronics and Mechanical Engineering, Kielce University of Technology, Al. 1000-lecia PP 7, 25-314 Kielce, Poland; tpala@tu.kielce.pl (T.P.); slipiec@tu.kielce.pl (S.L.); pkmid@tu.kielce.pl (I.D.)

**Keywords:** Al–Ti laminate, fracture, acoustic emission diagnostic, pattern recognition, clustering AE signal

## Abstract

The paper describes tests conducted to identify the mechanisms occurring during the fracture of single-edge notches loaded in three-point bending (SENB) specimens made from an Al–Ti laminate. The experimental tests were complemented with microstructural analyses of the specimens’ fracture surfaces and an in-depth analysis of acoustic emission (AE) signals. The paper presents the application of the AE method to identify fracture processes in the layered Al–Ti composite using a non-hierarchical method for clustering AE signals (k-means) and analyses using waveform time domain, waveform time domain (autocorrelation), fast Fourier transform (FFT Real) and waveform continuous wavelet based on the Morlet wavelet. These analyses made it possible to identify different fracture mechanisms in Al–Ti composites which is very significant to the assessment of the safety of structures made of this material.

## 1. Introduction

Design requirements imposed on contemporary structural members frequently make it necessary to use materials that combine different strengths and mechanical properties. Examples of such materials include composites, particularly laminates, which may consist of several layers of materials with different strengths and mechanical properties, selected depending on the specific needs of the user. Usually, one of the layers of laminate is responsible for structural strength, while the remaining layers may have special properties—corrosion protection, thermal insulation and sealing or damping of high-energy impact loads. The object under test described in this paper is an Al–Ti laminate consisting of three layers: outer lamella from an Al alloy (AA2519), inner lamella from an Al alloy (AA1050) and an outer lamella from a Ti alloy (Ti6Al4V).

Various techniques and technologies are used during production to join different metallic materials in the laminates—adhesive bonding, high-temperature welding, friction welding [1,2,3,4] and, increasingly, explosion welding [5,6,7,8,9]. This technology uses the explosive energy for mutual penetration of the bonded materials. The technology of explosion bonding is fairly new, and it is used to make various types of items consisting of different materials; in each specific case, the parameters of the process are selected based on experience with laboratory specimens. That is why, in order to use elements made of layered materials created by explosion welding, it is necessary to know their strength and mechanical properties. The least investigated and, at the same time, very important area, which frequently determines the strength of the layered material, is the interface of the bonded materials—lamellas. There are no theoretical studies concerning explosion bonding of different materials and mechanical characteristics of the created laminates, as a result of which the most reliable method is to assess their strength properties using experimental means [10,11,12,13,14,15]. On the other hand, the commonly applied methods of determining material characteristics have been prepared for homogeneous materials, which is why it is difficult to apply them directly to laminates, and, consequently, the determined physical and mechanical properties cannot be regarded as the actual material characteristics of the laminates.

The AE technique is commonly used to detect and monitor damage and development of such damage in various structures, and it is currently recognised as one of the most reliable and well-established methods of non-destructive testing (NDT) [16]. Acoustic emission is a very efficient and effective method of detecting cracking and fatigue of metals, glass fibre, wood, composite materials, ceramics, concrete and plastics [17,18,19]. It can also be used to detect faults and pressure leaks in tanks or pipes or monitor the progress of corrosion in welded joints [20].

Unlike other techniques, which can only detect geometric discontinuities, AE methods can detect fibre tearing, delamination of adjacent layers in laminated composite plates, matrix cracking and fibre pull-out [21]. Most AE signals are caused by friction or by friction among the damaged components of the composite. Potential application of the AE technique to the assessment of damage to composite materials was discussed in References [22,23], with the mechanisms behind the cracking of composite materials being described in Reference [23]. Studies of local damage to composite materials based on an analysis of the acoustic emission signal were carried out by Marec et al. [24]. The results of the tensile tests clearly identified damage mechanisms in various composite materials: cross-ply composites and sheet moulding compound (SMC). The tests indicated an evolution of damage in these materials over time until global failure, and they identified the most critical damage mechanisms. It was also found that the generated AE signals were subject to scattering and attenuation due to the elasticity of the waves. Most studies carried out in order to monitor AE signals in metals were focused on samples with the shape of thin plates [16,25]. However, the propagation of waves in thin plates is dispersive by definition [26] due to the different phase velocities at which different frequencies are propagated. Dang Hoang et al. [27] showed a relationship between the duration and energy of the AE signal and failures in aluminium plates joined together. In a similar study, the relationship between AE count rates and crack propagation rates in welded steel samples during fatigue was described in Reference [28]. Aggelis et al. [17] established that certain parameters of AE signals, such as the rise angle (RA), duration and rise time, are very sensitive to crack propagation rate, and can be useful in the characterisation of damage if the AE signals from rise time (RT) samples subject to fatigue are accordingly interpreted. The tests described in Reference [29] regarding the fatigue properties of steel and welds and fractographic and microstructural observations show that AE can be used as a tool to monitor damage caused by fatigue of the structure due to the fact of its sensitivity to changes within the crack.

That is why using AE in the study made it possible to identify certain peculiarities of the fracture process in the tested layered composite. Also, the AE signals were clustered depending on the processes that generated them using the iterative *k*-means method, which clusters AE parameters in a Euclidean space [30,31], and analysed them using waveform time domain, waveform time domain (autocorrelation), fast Fourier transform (FFT Real) and waveform continuous wavelet based on the Morlet wavelet [32].

Understanding the processes occurring in the contact areas of different materials has an important role in assessing laminate strength. Actually, these areas of contact of different layers are the most important in composites strength analysis. Theoretical and experimental studies on the effect on bonding strength are presented in References [33,34,35].

This paper presents the results of studies of the fracture process in Al–Ti laminate. To implement different fracture mechanisms the tests were carried out at two temperatures (i.e., *T*_1_ = 20 °C and *T*_2_ = −50 °C) on specimens with single-edge notches loaded in three-point bending (SENB). The force, specimens’ deflection, crack opening and acoustic emission (AE) signals were recorded during the loading. Based on mechanical sensors signals the fracture toughness characteristics were obtained and differences in loading diagrams were established. Observation of the specimens’ fracture surfaces using scanning electron microscope (SEM) showed some peculiarities of the cracking process of Al–Ti laminate in tested temperatures. Great attention was given to the analysis of AE signals to identify fracture mechanisms of Al–Ti laminate.

## 2. Material and Testing Methods

The object subject to laboratory tests was an Al–Ti laminate consisting of three layers: outer lamella from an Al alloy (AA2519), with a thickness of approximately 4.6 mm, inner lamella from an Al alloy (AA1050) with a thickness of approximately 0.2 mm, and an outer lamella from a Ti alloy (Ti6Al4V) with a thickness of approximately 4.6 mm. After the individual lamellas were bonded by explosion welding, the laminate was subject to heat treatment by soaking at 550 °C for 2 h, cooling to 165 °C and soaking at that temperature for 10 h. The authors of this paper analysed the fracture process in the laminate using specimens that had already been subject to welding and heat treatment. Information about the standard properties of base materials, welding process and heat treatment of the examined laminate was taken from References [12,15].

Experimental tests were carried out by the research team using SENB specimens (B = 10; W = 20; S = 80 mm) in three-point bending with a single-edge notch passing across all layers to a depth of 0.5W (Figure 1) at two temperatures: *T*_1_ = 20 °C and *T*_2_ = −50 °C. The strength characteristics of the component materials alloy at different temperature are given in Table 1. The specimens were prepared and loaded in accordance with ASTM E1820-09 [36]. During loading, the specimens were partially unloaded in order to determine the change in the compliance and calculate the growth of the crack. Tests at lowered temperature *T*_2_ = −50 °C were conducted in a thermal chamber in the environment of nitrogen vapours. During the tests, temperature variations did not exceed Δ*T* = ±1 °C. In order to determine fracture toughness characteristics, force *P*, deflection of the specimens (sensor 3) and crack opening displacement (*COD*) (sensor 4) were recorded while the load was applied to the sample. Acoustic emission signals were recorded using a 24 channel µSAMOS acoustic emission processor with the AEwin and NOESIS 12.0 software developed by the Physical Acoustics Corporation (PAC) from the Princeton, NJ, USA. Two low-frequency sensors with a flat response curve in the 30–80 kHz range (VS30-SIC-40dB, manufactured by Vallen GmbH) (sensor 5) and two broadband sensors with a frequency range of 100–1200 kHz (WD 100–1200 kHz, manufactured by PAC) (sensor 6) were used to record AE signals in a broad frequency range (Figure 1b).

AE parameters were clustered with the k-means method in a Euclidean space. This method involves the iterative search for the set of reference elements representing the individual clusters. Successive approximations of the reference elements are sought in each iteration through calculations using the indicated methods. Depending on the adopted assumptions, the reference element may be one of the elements of the *X* population or an element of a specific set *U**⊇*
*X*. In metric spaces, the reference element may be calculated as an arithmetic mean, and it accordingly represents the centre of gravity of the cluster. 

In general, the input for the k-clustering algorithm is the set of objects *X* and the expected number of clusters *k* and the output is the division into subsets *{C_1_, C_2_,..., C_k_}*. Usually, k-clustering algorithms belong to the category of optimisation algorithms. Optimisation algorithms assume that there is a loss function *k*: *{x|X*
*⊆*
*S} → R^+^* defined for every *S* subset. The purpose is to find the group with a minimal sum of the losses described by the following Formula (1): (1)$$\;E_{q}=\overset{k}{\underset{i=1}{\sum }}k\left(C_{i}\right)$$

In order to use the iterative algorithm, it is necessary to determine the dissimilarity measure used in the clustering process. In this case, it will be a point that is the resultant point for the particular cluster (representative element) calculated, for instance, as the geometric or arithmetic mean, described with the following Formula (2):(2)$$k(C_{i})=\overset{\left|C_{i}\right|}{\underset{r=1}{\sum }}d\left({\bar{x}}^{i},\;x_{r}^{i}\right)$$
where: ${\bar{x}}^{i}$- arithmetic (or geometric) mean of the cluster.

The most important advantage of this method is the speed of data processing and analysis.

The research also used other grouping methods (Forgy, Normalized Normal Constraint (NNC), Fuzzy C-Means (FCM), Gaussian Mixture Decomposition (GMD), Gustafson-Kessel (GK), Hidden Markov Model (HMM), Autoregressive Hidden Markov Model (ARHMM)) to isolate destructive subprocesses based on a full statistical-mathematical approach to divide into groups [37,38,39,40,41].

Analyses using waveform time domain, waveform time domain (autocorrelation), fast Fourier transform (FFT Real) and waveform continuous wavelet (CWT) based on the Morlet wavelet were also conducted in order to verify the divisions made using the k-means method. 

The waveform time domain chart shows a change of amplitude expressed in units of voltage V within a specific period. It is also useful in determining the rise time or duration of the acoustic signals for the indicated AE signal classes. The waveform time domain (autocorrelation) chart, in turn, is significant to technical diagnostics and to signal processing and transmission theory. The autocorrelation function is used to determine the rate of signal change and to detect periodic signals in “noisy” measurement signals—which is very important to the processing of diagnostic signals. The autocorrelation function $\varphi_{x}\left(\tau \right)$ of signal *x(t)* with limited energy is described by the following Formula (3): (3)$$\varphi_{X}\left(\tau \right)=\overset{\infty }{\underset{-\infty }{\int }}x\left(t\right)x^{0}\left(t-\tau \right)dt$$

The value of autocorrelation function $\varphi_{x}\left(\tau \right)\;$ at point *τ = 0* is real, and it is equal to the energy of the *x(t)* signal described with the following Formula (4): (4)$$\varphi_{X}\left(\tau \right)=\overset{\infty }{\underset{-\infty }{\int }}x{\left|x\left(t\right)\right|}^{2}dt=E_{x}$$

This property can be derived directly from Formula (3) after substituting *τ = 0*.

The charts of the fast Fourier transform (FFT Real) can be used for an analysis in the frequency domain to identify the key frequencies in the entire data set instead of examining every change in the time domain. The chart in the frequency domain shows the phase shift or strength of the signal at each frequency on which it exists. It shows how much signal is contained in the particular frequency band within a specific frequency range. 

Important frequency and energy-related information of AE signals can be extracted with the waveform continuous wavelet chart based on the Morlet wavelet. Waveform continuous wavelet (CWT) in the time domain is described by the following Formula (5):(5)$$CWT\left(t,a\right)=\frac{1}{\sqrt{\left|a\right|}}\overset{\infty }{\underset{-\infty }{\int }}x\left(\tau \right)\gamma \left(\frac{\tau -t}{a}\right)d\tau$$
where *t*—time instant of the tested signal*; x (τ)*—analysed signal; *γ (t)*—base filter function, so-called mother function; *a*—scale of the mother function.

The features extracted from this wavelet transform from can be significant to the detection of structural damage. By using a suitable family of the base signal filter function (so-called wavelet mother), it is possible to identify a temporary variation of the analysed signal. Another advantage of using *CWT* is the detection of the variation of the analysed value and/or—depending on the domain of the analysed response—identification of the location or time of damage. The Morlet wavelet was selected as the mother function for the analysis of the response signal obtained through the simulation of damage. The Morlet wavelet is currently used in various applications including successful use in damage diagnostics. 

## 3. Results of Experimental Tests

The critical values of fracture toughness were calculated as *J*-integral at the point of reaching the maximum force using the Rice formula: *J*_C_ = 2*A*_C_/*B*(*W* – *a*_0_), where *A*_C_ is the energy absorbed by plastic deformation and crack growth in the sample. For room temperature, *T*_1_ = 20 °C, the integral in accordance with the Rice formula reached *J*_C_ = 46 kN/m, and as the test temperature lowered, *T*_2_ = −50 °C, the critical value of *J*-integral decreased slightly, reaching *J*_C_ = 40 kN/m, which was consistent with expectations. For metals, particularly ferritic steels, a decrease of the test temperature will reduce critical fracture toughness characteristics [42,43,44,45,46].

However, careful observation and comparison of the *P*–*COD* load curves with the recorded AE signals indicates certain peculiarities of the fracture process in Al–Ti laminate specimens tested at two temperatures: *T*_1_ = 20 °C and *T*_2_ = −50 °C (Figure 2). For the lowered temperature, the *P*–*COD* curve has slightly lower values than for room temperature. It should be emphasised that in samples made from homogeneous materials, Al (AA2519) and Ti (Ti6Al4V) alloys [15], and in specimens from steel [15,47], an opposite trend could be observed—reducing the temperature increased the strength characteristics and Young’s modulus, and it made the rise of the chart in the loading section accordingly steeper. Also, in the rising sections of the load curves, we can observe a deviation from linearity occurring when the force is approximately 4 kN for the specimens tested at temperature *T*_2_ = −50 °C and approximately 6 kN for tests at temperature *T*_1_ = 20 °C.

The distributions of AE signals in both samples (Figure 2) clearly show that signals with a high strength (above 5.0 × 10^7^ pV∙s) appear when the force is decreasing, directly after reaching the maximum force levels which corresponds to the start of the main subcritical crack. Also, the AE signals with high strength for the sample tested at *T*_2_ = −50 °C were located primarily in short time sections just after the maximum force level, which indicates a rapid growth of the subcritical crack. For the sample tested at *T*_1_ = 20 °C, AE signals with high strength also appeared while the load was being applied which indicates a continuous development of the subcritical crack.

AE signals for which signal strength (SS) was within the range of 1.0 × 10^7^–5.0 × 10^7^ pV∙s were very significant to the assessment of the generated destructive processes in the tested samples. These signals appeared in various phases of the loading of the samples tested at *T*_2_ = −50 °C and *T*_1_ = 20 °C. At room temperature, most of these signals were recorded along the falling part of the *P*–*COD* load curve, representing the increase of subcritical crack. However, during the test at lowered temperature (*T*_2_ = −50 °C), the great majority of signals from that range were recorded in the specimen loading phase before the maximum force was reached, i.e., when the subcritical crack did not start yet. The observed difference in the appearance of signals from that range may indicate that the Al–Ti laminate fracture process develops differently at different temperatures. This assumption is also supported by the peculiarities observed in the *P*–*COD* curves as indicated above.

Fracture surfaces of the samples were observed with a scanning electron microscope (SEM), and the recorded AE signals were analysed using a non-hierarchical method for AE signal clustering based on *k*-means [30,31] and analysis using waveform time domain, waveform time domain (autocorrelation), fast Fourier transform (FFT Real) and waveform continuous wavelet based on the Morlet wavelet [32] in order to explain the peculiarities of the *P*–*COD* curves at different temperatures and identify the mechanisms generating the AE signals. 

## 4. Tests of Fracture Surfaces and Microstructure Using SEM

Observations of the fracture surfaces of the tested specimens using an optical microscope and SEM clearly indicated the formation and development of two mutually perpendicular cracks—the main crack, which developed from the pre-crack in the same plane, and the delamination crack, which developed in a perpendicular plane between the component layers of Ti and Al alloys (Figure 3). The presence of delamination cracks was observed in all samples tested at both temperatures. In most cases, the delamination crack was formed on the Ti6Al4V alloy side, but there were also situations where the crack was formed at the AA2519 alloy layer, or even in parallel on both sides next to the base layers (Figure 3b).

The nature of the delamination crack clearly indicates the presence of two mechanisms in the development of the fracture—brittle phase fracture and shear in the AA1050 aluminium alloy (Figure 4). Most likely, a brittle fracture may be initiated by the brittle fracture of the particles of Al and Ti intermetallic compounds [45] or oxides formed in the transition zone (TZ) between the Al alloy (AA1050) and Ti alloy (Ti6Al4V) during explosion welding (Figure 5). The presence of elements of metals and oxygen was confirmed through EDS analysis in the transition zone (Figure 6, Table 2). In the photographs (Figure 4), the areas of the fractured brittle phases are clearly visible as the light areas from which shear develops in the Al alloy (AA1050). In samples tested at temperature *T*_1_ = 20 °C, the growth of the delamination crack occurred primarily by shear (Figure 4a), and at the reduced temperature, *T*_2_ = −50 °C, there were large areas with brittle fracture (Figure 4b).

Observations of metallographic photographs of the connecting zone of the base materials (Figure 5) indicated that this narrow strip with a width of up to 40 μm (transition zone—TZ) between the material of the inner layer, AA1050 alloy, and the Ti6Al4V alloy (Figure 5b), was the weakest link in the Al–Ti laminate. This is precisely the strip formed by the mutual penetration of materials during explosion bonding. The appearance of brittle fracture in that zone is caused by different types of particles and material discontinuities present in the TZ.

The complementary tests conducted using the EDS analysis showed that the TZ contained elements of metals from the base layers (Figure 6a), which formed Al and Ti intermetallic compounds [47], and oxygen (Figure 6b), which indicated the existence of metal oxides (Table 2).

The development of the main crack, which grew from the pre-crack in the same plane, occurred in both alloys: Al (AA2519) and Ti (Ti6Al4V), in accordance with the ductile mechanism through the formation and connection of voids. However, there are certain noticeable differences. In the Al alloy (AA2519), voids with a size of 10–20 μm are formed around fairly large particles with a size of 2–5 μm (Figure 7a). The morphology of the fracture surface of the Ti (Ti6Al4V) alloy, in turn, consists of smaller fragments of voids with a size of 2–7 µm (Figure 7b). The nature of the propagation of the main crack is similar in the samples tested at both temperatures.

## 5. Analysis of AE Signals

The AE signals recorded during loading of the specimens at *T*_1_ = 20 °C and at *T*_2_ = −50 °C (Figure 2) were preliminarily clustered using the non-hierarchical method for AE signal clustering—k-means. As a result of the clustering, the signals were divided into five classes shown on point charts of AE signal strength (SS) pV∙s, over time (s) (Figure 8).

The charts also show the force acting on the sample, P (kN), which was recorded with a set of acoustic instruments only when the AE signals were present. Since the sample was loaded and partially unloaded, the AE signals intensify when the load is rising and diminish as the load is decreasing.

It is to notice that in the sample tested at *T*_1_ = 20 °C (Figure 8a), the great majority of AE signals from classes 1–5 occurred just after the maximum force was reached and later, i.e. after the main crack has propagated. When the load was rising, only AE signals with low signal strength (SS) were recorded—for class 1, the SS reached 8.0 × 10^6^ pV∙s, and, for class 2, the SS values were within the range of 4.8 × 10^6^ to 2.4 × 10^7^ pV∙s.

During tests at temperature *T*_2_ = −50 °C (Figure 8b), in the rising part of the load curve, up to the point at which maximum force was reached, numerous signals of classes 1 to 4 were recorded, which indicates an intense growth of the crack during that period. Just after the maximum force was reached, AE signals of classes 1 to 5 were recorded, with no signals of classes 2–5 being recorded after the force dropped below 0.68 P_max_ (less than 5.3 kN). 

The preliminary clustering of AE signals using the k-means method was followed by an in-depth analysis of AE signals using the following charts: waveform time domain (WTD) (Figure 9), fast Fourier transform (FFT Real) and waveform continuous wavelet using the Morlet wavelet (Figure 10, Figure 11, Figure 12, Figure 13 and Figure 14) and waveform time domain (autocorrelation)—complex Fourier series (Figure 15).

The preliminary clustering of AE signals using the *k*-means method was followed by an in-depth analysis of AE signals using the following charts: (WTD) (Figure 9), waveform frequency domain (Real)—FFT Real and waveform continuous wavelet using the Morlet wavelet (Figure 10, Figure 11, Figure 12, Figure 13 and Figure 14) and waveform time domain (autocorrelation)—complex Fourier series (Figure 15). The charts shown in Figure 9, Figure 10, Figure 11, Figure 12, Figure 13, Figure 14 and Figure 15 can be used to verify and adjust the preliminary clustering of AE signals. In Table 3 are presented ranges and maximum levels of AE signal characteristics obtained by this analysis.

Class-5 signals (red points) are generated in connection with the ductile development of the main crack by the growth of voids, which is driven by the dislocation motion. They are characterised by a high amplitude of up to 8.81 V and low frequency level in the two dominant bands (Figure 14a,b): 60 and 70 kHz. The 60 kHz frequency is characteristic to aluminium alloys, and the frequency of 75 kHz characterises titanium alloys. These processes related to the growth of voids have long durations of up to 4000 μs (Figure 9e), high energy values of up to 12,000 eu (Figure 15e) and signal strength from 8.8 × 10^7^ to 1.2 × 10^8^ pV∙s (Table 3).

Class-4 signals (Bottle Green points) also include two characteristic frequency ranges: approx. 60 kHz and 250–300 kHz. The structure of the class-4 signal indicates that it is generated in the shear process, plastic deformations appear first with a frequency of approximately 60 kHz, followed by shear fracture characterised by a frequency of 250–270 kHz (Figure 13a,b). These signals are accompanied by a high amplitude, reaching 8.8 V and duration of up to 1000 μs (Figure 9d). The signal strength for class-4 signal ranges from 3.0 × 10^7^ to 5.8 × 10^7^ pV∙s, and the signal energy calculated from the WTD reaches up to 18,000 eu (Figure 15d). 

Class-2 (dark blue) and class-3 (pink) AE signals are characterised by a similar frequency in the range of 250–270 kHz and duration of approximately 520 μs, but they differ in amplitude, signal strength and energy (Table 3).

A higher amplitude, reaching up to 8.8 V could be observed with class-3 signals. These signals reached signal strength in the range from 1.1 × 10^7^ to 4.6 × 10^7^ pV∙s, and their maximum energy calculated from the WTD was 11,500 eu (Figure 9c, Figure 12a,b and Figure 15c).

Class-2 signals were characterised by lower signal strength, in the range from 4.8 × 10^6^ to 2.4 × 10^7^ pV∙s and 3.5 times lower energy, not exceeding 3000 eu (Figure 9b, Figure 11a,b and Figure 15b), determined through the WTD analysis. Both types of signals were generated by the brittle fracture mechanism. Most likely, they characterise brittle phase fracture in the transition zone: class-3 signals—fracture of Al–Ti intermetallic compounds, whereas class-2 signals indicate oxide fracture.

Class-1 (Light green) AE signals with a frequency of approximately 100 kHz had a very low amplitude, not exceeding 0.4 V, and they were characterised by signal strength in the range from 1.8 × 10^4^ to 8.0 × 10^6^ pV∙s and energy of up to 10 eu determined through the WTD analysis (Table 3, Figure 9a, Figure 10a,b and Figure 15a). 

These signals were generated by “noise”, and they were usually caused by the friction of the rollers against the sample or by the operation of the strength testing machine. Class-1 signals will not be considered in further analysis of the fracture process.

## 6. Discussion

Based on the test results discussed above in this paper, we will now try to describe the fracture process in the Al–Ti laminate. During the test at temperature *T*_1_ = 20 °C, the recorded signals of AE indicated that when loading the specimens up to *P*_max_ only brittle phase fracture, most likely fracture of oxides, occurred in the transition zone between the AA1050 alloy and the Ti6Al4V alloy, which were well reflects by class-2 AE signals. At the time when value of force *P*_max_ was reached, fracture occurred in various components of the Al–Ti laminate according to different mechanisms. Brittle fracture of Al and Ti intermetallic particles occurs in the transition zone. This process was accurately illustrated by class-3 signals. Also, this was the time of the appearance of shear fracture, i.e. delamination crack, characterised by class-4 signals. Almost simultaneously to this process, in the base layers of the AA2519 and Ti6Al4V alloys, the main crack also developed due to the growth of voids, which was described by the class-5 signals. The growth of the delamination crack and the main crack decreases the force because the sample is loaded by displacement. 

According to our results, the fracture process was slightly different in samples tested at *T*_2_ = −50 °C. AE signals of classes 2, 3 and 4 could be observed already at early stages of sample loading, before value *P*_max_ is reached. The signals were generated because the brittle fracture toughness of the materials was reduced along with the decrease in temperature [42,43,44,45,46]. That was why at *T*_2_ = −50 °C brittle phase fracture in the transition zone occurred at a lower load (i.e., classes 2 and 3) which, in turn, led to the fracture caused by shear in the connecting layer of the AA1050 alloy (class 4). This means that the delamination crack was present already when the main crack started to appear. This conclusion was confirmed by the results of mechanical tests, which indicated that the compliance of the specimen has decreased. When the specimen reaches its maximum force, *P*_max_, the main crack was propagated by the ductile mechanism of the growth of voids (class 5) and followed by subsequent stages of fracture by other mechanisms.

In a sense, the results presented in this paper are confirmed by Reference [48], where the authors described the evolution of the waves of AE signals along with the increase of the fatigue crack. It was found that the change of the shape of the AE wave was closely related to the physical condition of fracture loading and to the mechanism behind the growth of fatigue cracks. It was demonstrated that the load level could be associated with the acoustic emission signals present during the growth of fatigue crack.

The results presented in this paper are consistent with the results of fracture tests in metallic layered composites prepared by adhesive bonding [49,50,51], which indicate that different stages of the fracture process are represented by AE signals with different characteristic parameters.

Comprehensive and in-depth analysis of AE signals confirmed occurring various mechanisms of failure in the cracking process of the Al–Ti composite, which tested in the range of ambient to cryogenic temperatures [52].

## 7. Conclusions

This paper presents a comprehensive study of the fracture process in an Al–Ti laminate prepared by explosion welding. The testing methods used are modern and the results obtained are novelty. Mechanical load curves of SENB samples, *P–COD*, were examined in order to accurately interpret the fracture process; the fracture surfaces of the samples were thoroughly tested using SEM, and AE signals were recorded and subjected to various methods of analysis of wave packets of AE signals. 

Based on the loading tests of the SENB specimens at *T*_1_ = 20 °C and *T*_2_ = −50 °C temperatures and the load curves, it was found that the *P–COD* curves had an atypical form, particularly at lowered temperatures. 

Tests of the fracture surfaces of the specimens using optical and SEM microscopes found a delamination crack, which is usually initiated by the brittle phase fracture in the transition zone between the inner layer of the AA1050 alloy and the layer of the Ti6Al4V alloy. Then, the delamination crack develops primarily through the shear fracture mechanism. The propagation of the main crack in the base layers of the AA2519 and Ti6Al4V alloys occurs through the ductile mechanism of the growth of voids for two test temperatures.

The recording of AE signals and using different methods for the analysis of these signals, including non-hierarchical clustering methods (*k*-means) and analyses using Waveform Time Domain, Fast Fourier Transform (FFT Real), Waveform Continuous Wavelet using the Morlet wavelet and Waveform Time Domain (Autocorrelation), enabled the identification of four classes of signals and characterisation of the primary mechanisms of the fracture processes in the tested layered Al–Ti composite:Class 2—AE signals of the brittle phase fracture of Al and Ti oxides;Class 3—AE signals of the brittle phase fracture of Al and Ti intermetallic compounds;Class 4—AE signals generated during the formation of the delamination crack through shear in the Al (AA1050) layer;Class 5—AE signals generated during the development of the main crack in the base material layers, Al (AA2519) and Ti (Ti6Al4V) alloys.

The recorded AE signals were used to determine the primary differences in the fracture process of the Al–Ti laminate. At temperature *T*_1_= 20 °C, the growth of the delamination crack and the main crack occur almost simultaneously. However, at temperature *T*_2_ = −50 °C, the delamination crack precedes the main crack, and the development of cracks in the base materials occurs without interaction between them. This explains the non-characteristic and illogical behaviours observe in the specimen load curves (*P–COD*) at different test temperatures. 

These comprehensive tests indicated that the methods of analysing AE signals can be effectively used to identify the development of cracks in structural members. They can identify characteristic mechanisms of the formation and development of defects, also at very early stages of damage evolution, which cannot be achieved with other NDT methods. This fact can be used to create an automatic diagnostic system capable of determining the types and mechanisms of the development of potential damage at every stage of material use and assessing the reliability of the structure.

## Figures and Tables

**Figure 1 materials-13-02909-f001:**
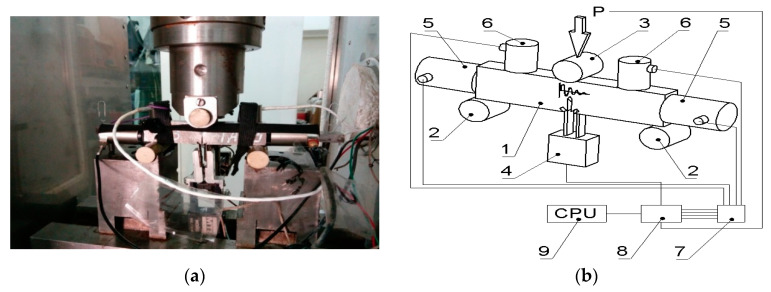
The SENB specimen in three-point bending with installed AE sensors: (**a**) photograph of the specimen in the thermal chamber; (**b**) scheme of loading and installation of mechanical and AE sensors.

**Figure 2 materials-13-02909-f002:**
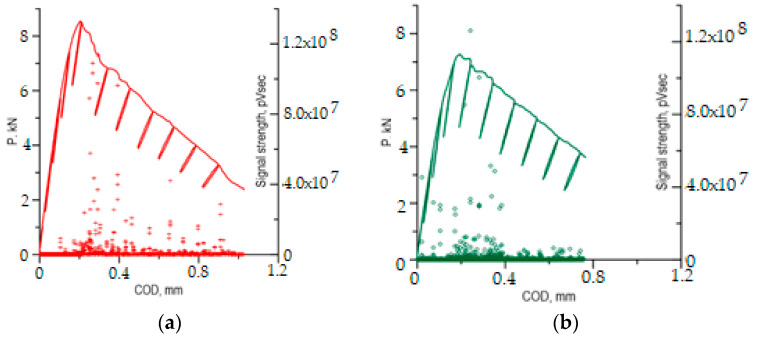
Load curves of the specimens with AE signals. (**a**) *T*_1_ = 20 °C; (**b**) *T*_2_ = −50 °C.

**Figure 3 materials-13-02909-f003:**
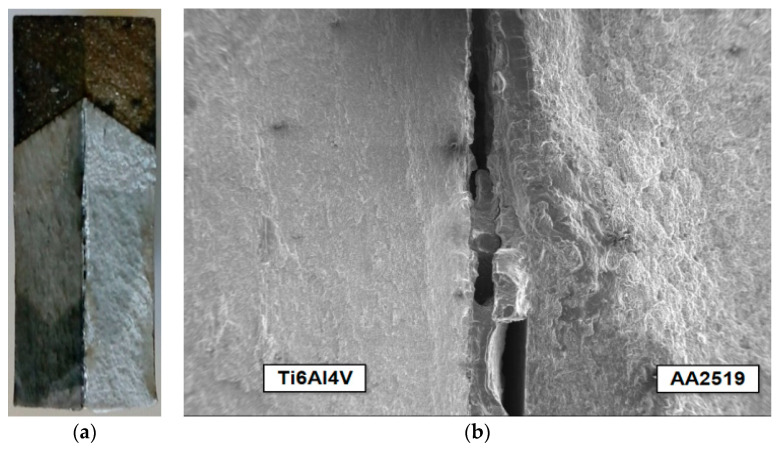
Delamination crack: (**a**) total view of fracture surface of the SENB specimen; (**b**) development of the delamination crack.

**Figure 4 materials-13-02909-f004:**
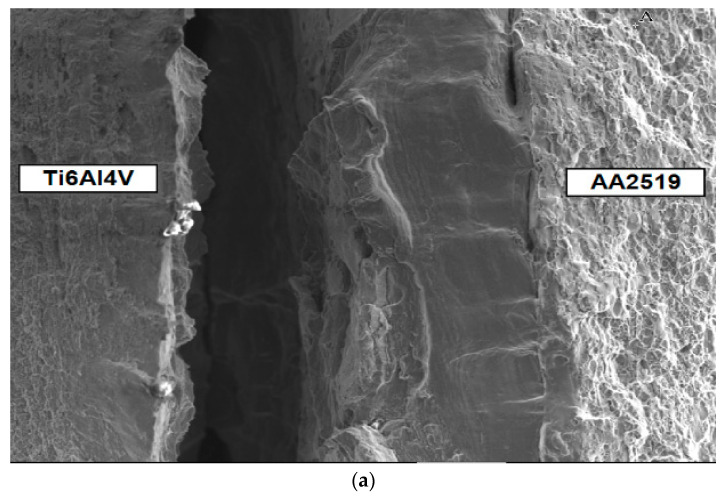
Nature of the delamination crack. (**a**) tests at *T*_1_ = 20 °C; (**b**) tests at *T*_2_ = −50 °C.

**Figure 5 materials-13-02909-f005:**
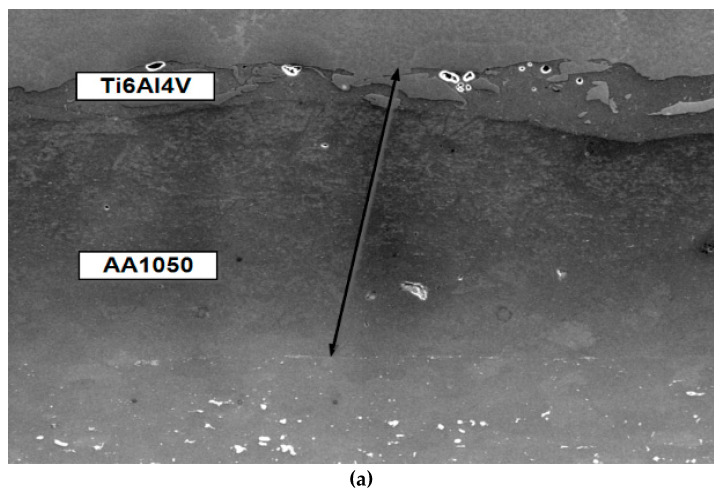
(**a**) inner layer and connecting zone of the base materials; (**b**) transition zone (TZ) between the AA1050 alloy and the Ti6Al4V alloy.

**Figure 6 materials-13-02909-f006:**
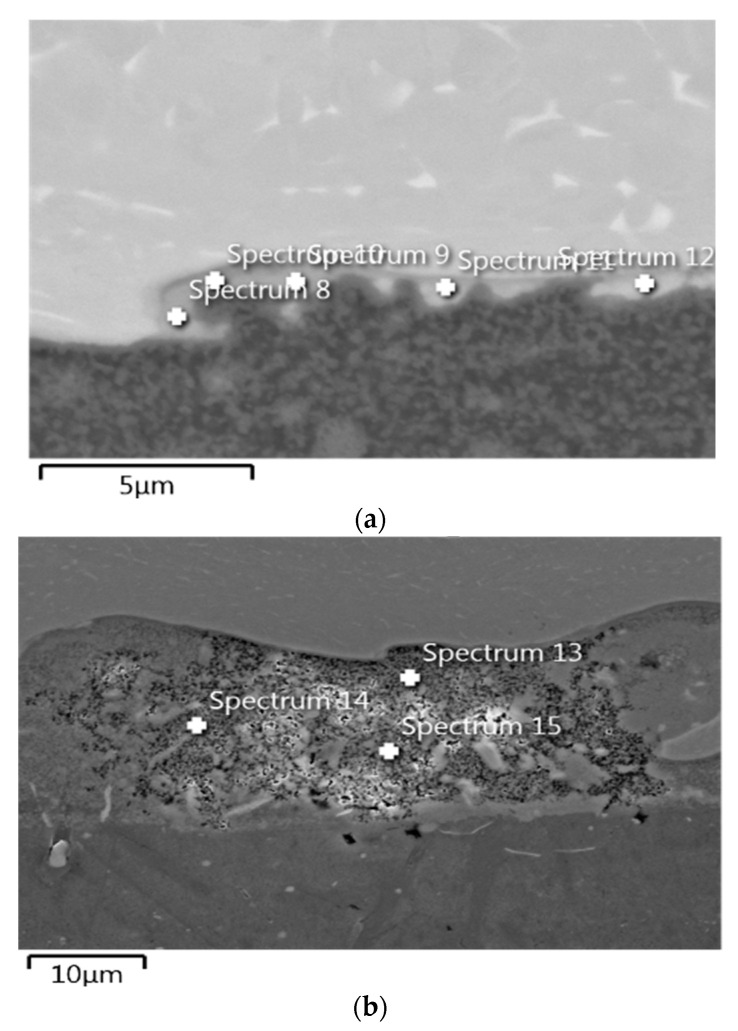
The point of EDS analysis in TZ in aria 1 (**a**) and aria 2 (**b**).

**Figure 7 materials-13-02909-f007:**
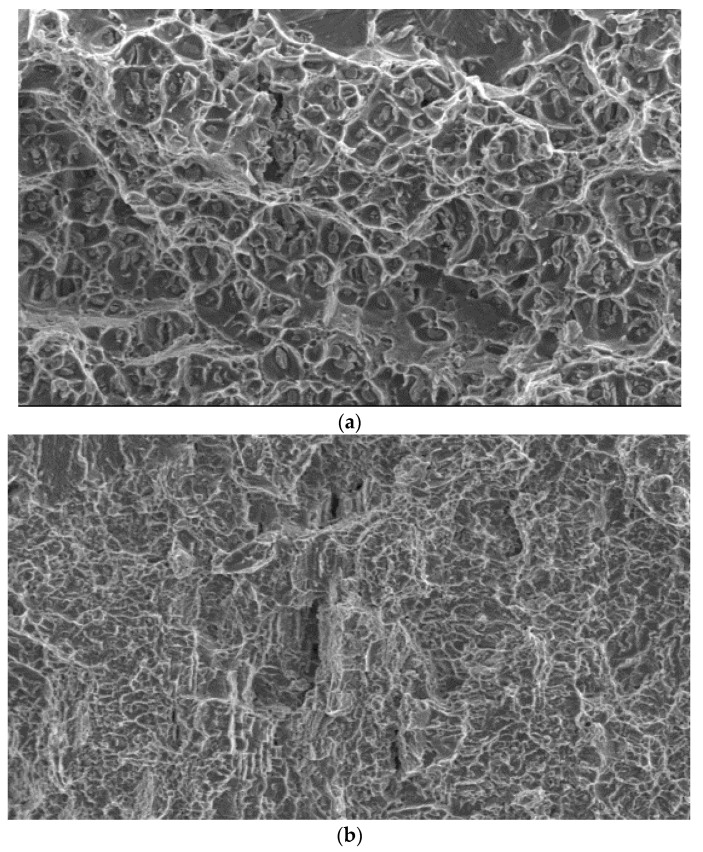
Fracture mechanism in the base materials. (**a**) Al alloy (AA2519); (**b**) Ti alloy (Ti6Al4V).

**Figure 8 materials-13-02909-f008:**
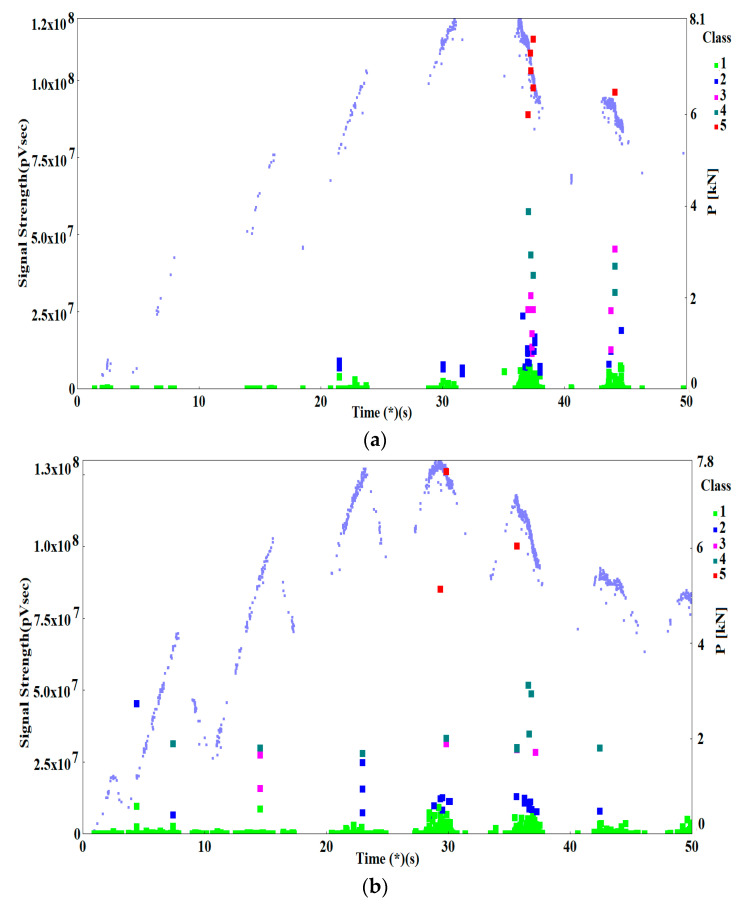
Point charts of signal strength (SS) over time with breakdown into classes describing the characteristic fracture mechanisms. (**a**) for *T*_1_ = 20 °C; (**b**) for *T*_2_ = −50 °C.

**Figure 9 materials-13-02909-f009:**
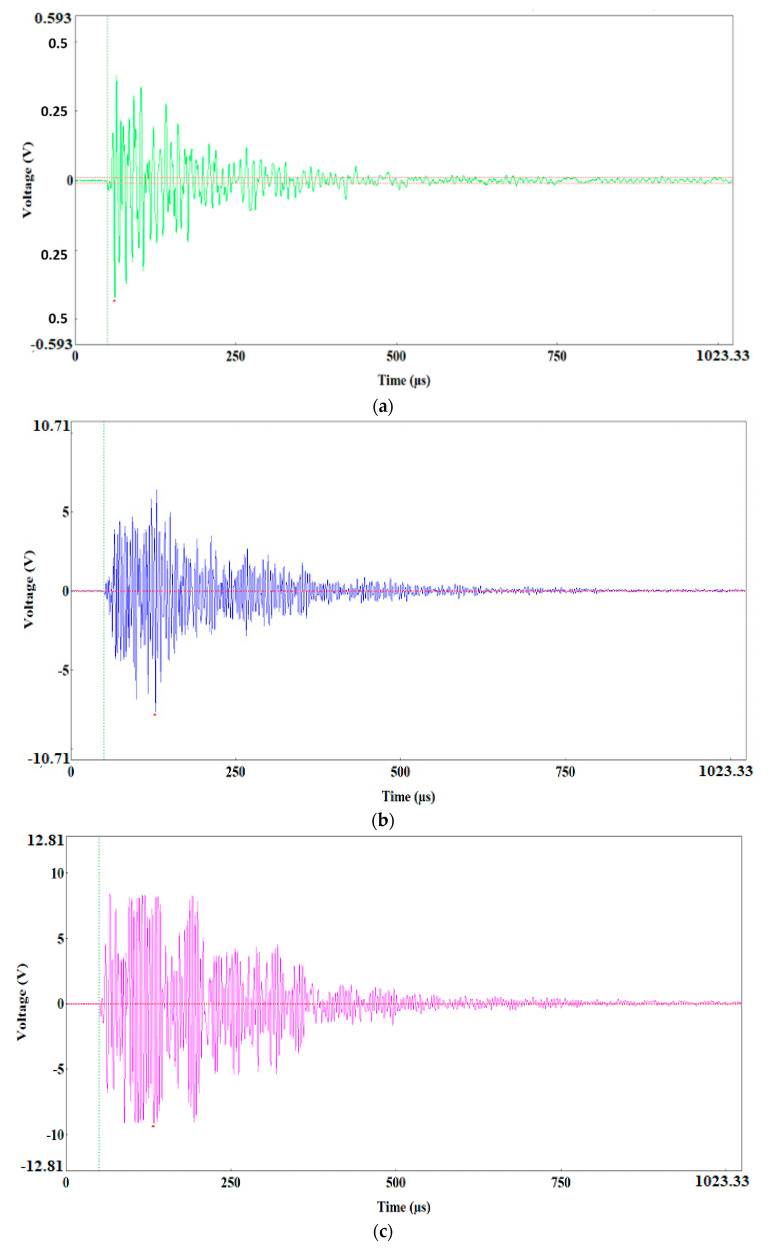
Waveform time domain charts for the individual classes of AE signals. (**a**) class 1; (**b**) class 2; (**c**) class 3; (**d**) class 4; (**e**) class 5.

**Figure 10 materials-13-02909-f010:**
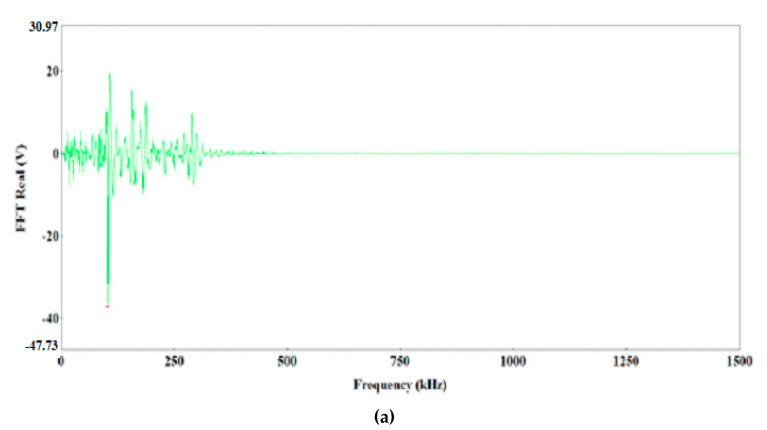
Waveform frequency domain (Real) (**a**) and waveform continuous wavelet (Morlet) (**b**) charts for the individual classes of AE signals (class 1).

**Figure 11 materials-13-02909-f011:**
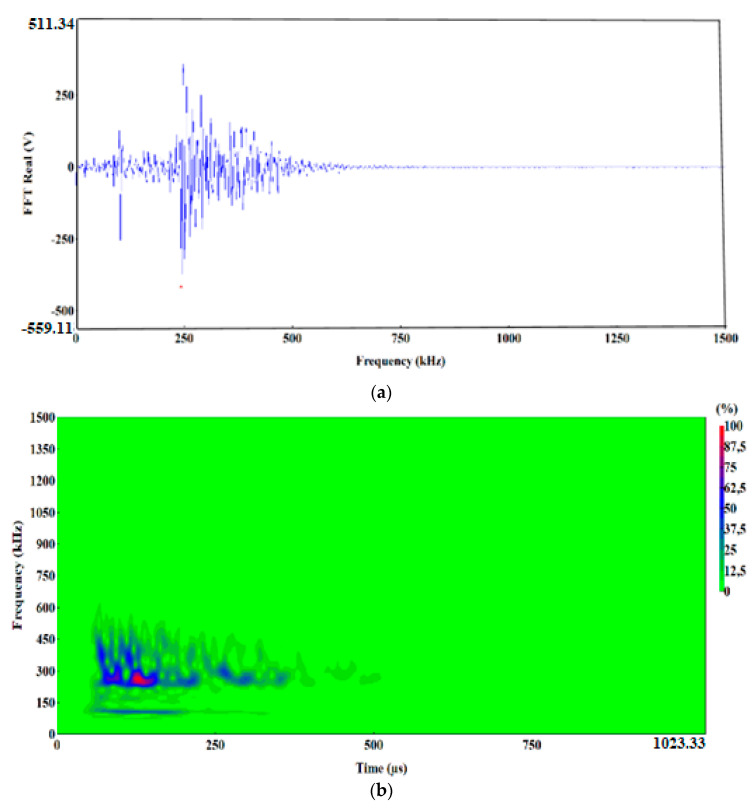
Waveform frequency domain (Real) (**a**) and waveform continuous wavelet (Morlet) (**b**) charts for the individual classes of AE signals (class 2).

**Figure 12 materials-13-02909-f012:**
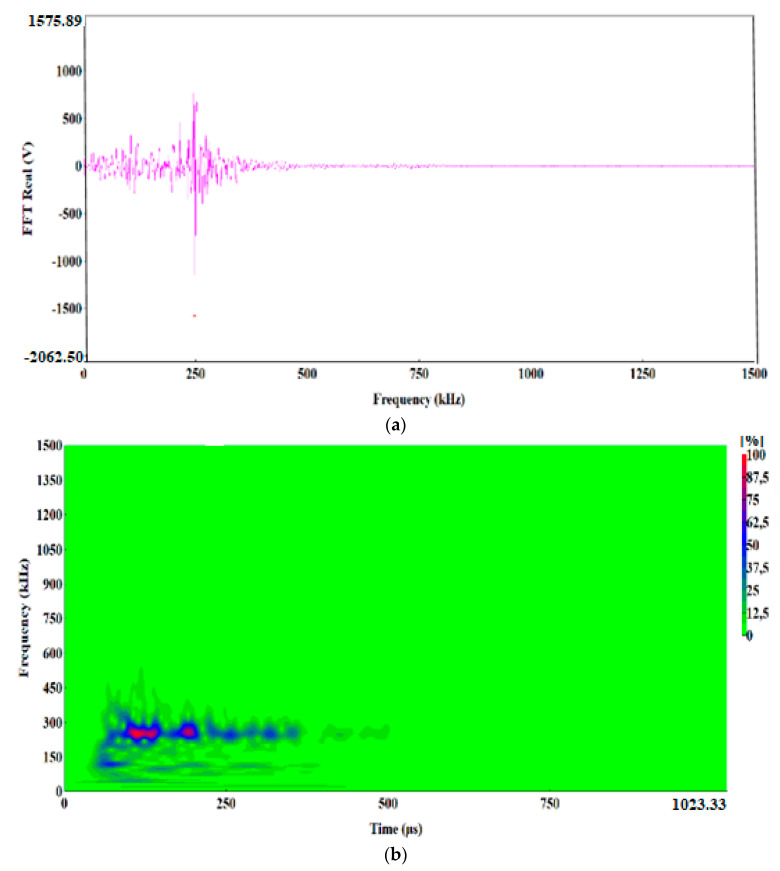
Waveform drequency domain (Real) (**a**) and waveform continuous wavelet (Morlet) (**b**) charts for the individual classes of AE signals (class 3).

**Figure 13 materials-13-02909-f013:**
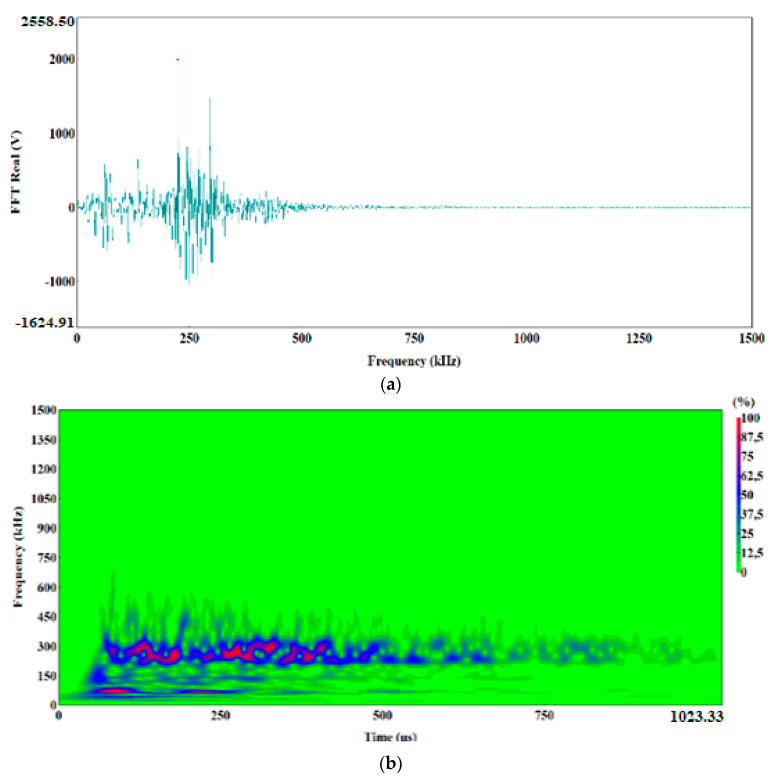
Waveform frequency domain (Real) (**a**) and waveform continuous wavelet (Morlet) (**b**) charts for the individual classes of AE signals (class 4).

**Figure 14 materials-13-02909-f014:**
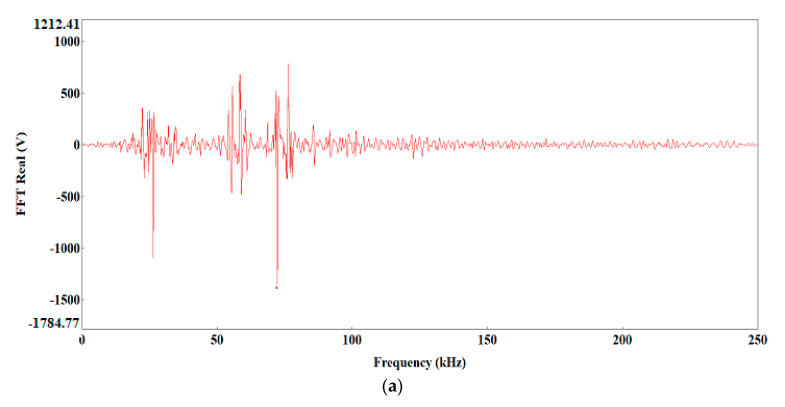
Waveform frequency domain (Real) (**a**) and waveform continuous wavelet (Morlet) (**b**) charts for the individual classes of AE signals (class 5).

**Figure 15 materials-13-02909-f015:**
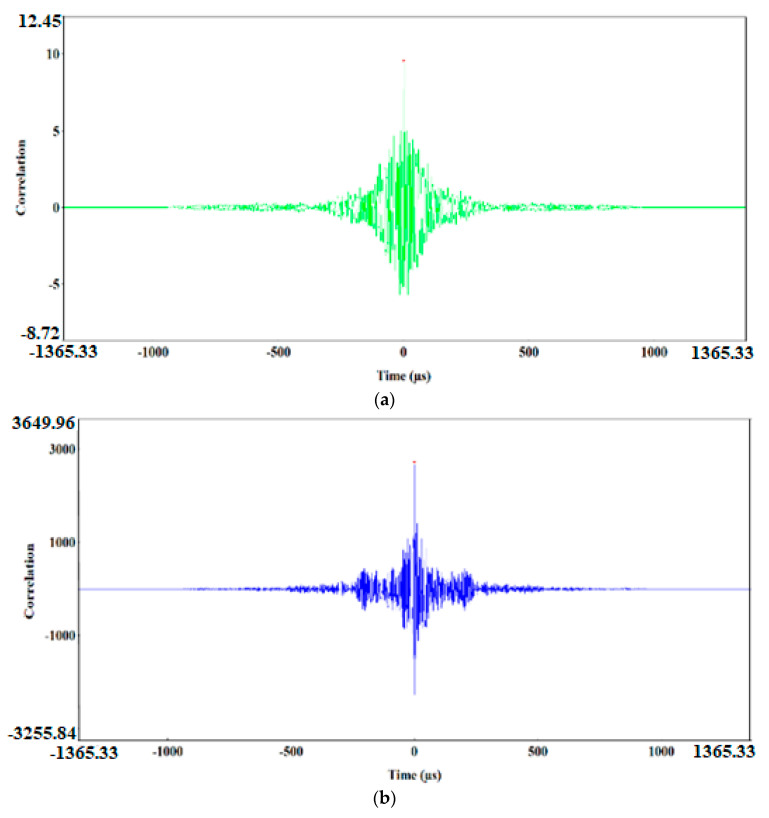
Waveform time domain (autocorrelation) chart for the individual classes of AE signals. (**a**) class 1; (**b**) class 2; (**c**) class 3; (**d**) class 4; (**e**) class 5.

**Table 1 materials-13-02909-t001:** Mean values of the mechanical properties of AA2519 aluminium alloy and Ti6Al4V titanium alloy at different temperature.

Material	Temperature	σ_y_	σ_u_	E	A_5_
	(°C)	(MPa)	(MPa)	(GPa)	(%)
**AA2519**	20	301	560	67.8	16.3
	−50	320	607	72.8	16.9
**AA1050**	20	150	194	69.2	18.2
	−50	161	208	74.3	18.5
**Ti6Al4V**	20	859	908	111.7	13.6
	−50	937	1140	117.8	13.8

**Table 2 materials-13-02909-t002:** The results of EDS analysis.

Result Type	Weight %					
Spectrum N	O	Al	Si	Ti	V	Total
**Spectrum 8**	-	32.90		64.53	2.57	100.00
**Spectrum 9**	-	33.22	0.38	63.57	2.83	100.00
**Spectrum 10**	-	32.35	0.32	64.66	2.67	100.00
**Spectrum 11**	-	30.80	0.33	66.19	2.69	100.00
**Spectrum 12**	-	26.15	0.30	70.47	3.08	100.00
**Spectrum 13**	13.42	48.56	7.63	28.93	1.46	100.00
**Spectrum 14**	14.44	51.46	8.07	24.90	1.14	100.00
**Spectrum 15**	15.97	46.82	10.27	25.89	1.04	100.00

**Table 3 materials-13-02909-t003:** Ranges and maximum levels of AE signal characteristics.

Class	1	2	3	4	5
**Signal Strength (SS) (pV∙s)**	1.8 × 10^4^ ÷ 8.0 × 10^6^	4.8 × 10^6^ ÷ 2.4 × 10^7^	1.0 × 10^7^ ÷ 3.0 × 10^7^	3.0 × 10^7^ ÷ 5.8 × 10^7^	8.8 × 10^7^ ÷ 1.2 × 10^8^
**Amplitude (V)**	0.43 (4%)	7.5 (75%)	8.8 (80%)	8.8 (88%)	8.81 (88%)
**FFT Real (V)**	±(20 ÷ 38)	±350	±1000	±(1000 ÷ 2000)	±(750 ÷ 1500)
**Frequency (kHz)**	100	250 ÷ 270	250 ÷ 270	60; 250 ÷ 300	60; 75
**Duration (μs)**	400	520	520	1000	4000
**WTD (Auto Correlation) Energy (eu)**	10	3000	11500	18000	12000

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
