# Peer review of "Using AE Signals to Investigate the Fracture Process in an Al–Ti Laminate"

_materials, 2020, doi:10.3390/ma13132909_

Round 1

Reviewer 1 Report

Introduction chapter. The novelty of the work is not clarified and literature review does not support this goal.

Line 12 Abbreviations should be avoided in the summary, or used with explanation (single edge notched loaded in three-point bending (SENB)).

The quality of Figures (3-12) needs to be improved in order to clear the numeric and text information.

Organization of the manuscript should be improved. It is necessary to clearly describe the structure of the article in the introduction.

Results. Figures are very helpful, but where is the comparison with existing literature? In addition, where is the critical analysis of results? Could authors deeply analyze the Fig. 9-12, its difficult understand what the main idea authors want to present. Figures 9-12 are presented at the end of the article, the results obtained should be described in more detail case and the analysis should be presented.

The Figures shouldn‘t be in the end of the Chapters in an article (Figure 12).

Figures 10 and 11 should be merged.

Conclusions. Authors should write conclusion focused in the novelty of getting results. In my opinion it’s not necessary to write one more abstract in the conclusion chapter. Authors must modified it accord this remark.

Author Response

Thank you very much for your valuable comments and assessment of our article.
We introduce some additional information into paper text according to Reviewers comments

Reviewer 2 Report

This study proposed a methodology to investigate the fracture phenomena of an Al-Ti laminate. This is an important research field and the paper presents an interesting contribution for better understanding the fracture process of a three-layer Al-Ti laminate. Only a few minor key point.

Theoretical studies on the bounding (or interface) effect of laminates are available in many literatures. This should be included in the “introduction” section. For example,

  1. J. Kong, C. Ruan-Wu, Y. X. Luo, C. L. Zhang, Ch. Zhang. 2017. “Magnetoelectric effects in multiferroic laminated plates with imperfect interfaces,” Theoretical and Applied Mechanics Letters, 7:93-99.
  2. Dinghe Li, Yan Liu. 2012. “Three-dimensional semi-analytical model for the static response and sensitivity analysis of the composite stiffened laminated plate with interfacial imperfections,” Composite Structures: 94:1943-1958.
  3. Jun-Sik Kim, Jinho Oh, Maenghyo Cho. 2011. “Efficient analysis of laminated composite and sandwich plates with interfacial imperfections,” Composites Part B: Engineering, 42:1066-1075.

By addressing the above reviewer’s concern, the manuscript should be good for publication. Therefore, a minor revision is recommended.

Author Response

Thank you very much for your valuable comments and assessment of our article

Reviewer 3 Report

This paper study the fracture process in the Al-Ti laminate.

Q1. Line 62: "Most AE signals are caused by friction or by friction between the damaged components of the composite. " Are you sure of that? Does it depend on the loading profile? on the composite layup? laminated or not?

Q2. "The tests indicated an evolution of damage in these materials over time until global failure, and they identified the most critical damage mechanisms": be careful about the type of clustering method used (see comments below).

Q3. You can complete Ref 17 with ref [] which shows the influence of damage accumulation during fatigue of composite (ring & laminate) on most of current AE features.

Can you complete the literature with more recent papers and different approaches for clustering. You mention kmeans but it is the simplest approach for clustering. Other approaches have been proposed
- FCM [REFX1] : https://ieeexplore.ieee.org/document/1198989
- GK [REFX2] : https://ieeexplore.ieee.org/document/7166324
- HMM [REFX3] : https://www.tandfonline.com/doi/abs/10.1080/10910340600996175

Can you at least mention those alternative works and give a few arguments for kmeans?

Q4. Your formulation of kmeans page 5 is not standard.
In eq1. there is an index "k" in the sum, and "k" within the sum. This is not clear. I do not see the number of points, it is not normal. Generally the objective is sum_n sum_k r(nk) d(xn, ck) where xn is a feature vector in R^d (as many as AE transients detected), ck is cluster k, rnk is in {0,1} for kmeans.

Q5. The Waveform Time Domain definition and surrounding text (lines 142-151) are not clear. You mean you compute the autocorrelation for each AE signal? So you get only the maximum that is used as a feature? Please improve this part.

Can you justify the statement "can be used to identify the synchronisation signal required to receive the transmitted information from the combination of the information signal and interference"? How is it useful?

The same for "The autocorrelation function is used to determine the rate of signal change and to detect periodic signals in “noisy” measurement signals"?

If you just compute an energy from it then just use eq 4 and remove the text that is not clear and useless to understand this part.

Q6. Eq 4 is weird:
- tau equals 0 isn't it?
- and x|x(t)|^2 means that x is at power 3? so it is not really an energy?

Q7. Line 160: "Important information about the properties of signals clustered using the k-means method is shown in the Waveform Continuous Wavelet chart based on the Morlet wavelet."

I think this sentence should be reformulated into "Important frequency and energy-related information of AE signals can be extracted with the Waveform Continuous Wavelet chart based on the Morlet wavelet." because to my opinion it is independent on the clustering method.

Q8. The statement "The wavelet transform is significant to the detection of structural damage." is not clear. What do you mean? that the features extracted from this transform can be relevant for detecting damages? In that case which features did you use? The wavelet transform creates a 3D map, is that used for clustering?

Q9. A table with the list of features used is mandatory.

Q10. How did you find the number of clusters?

Q10. "the recorded AE signals were analysed using a non-hierarchical method for AE signal clustering
based on k-means": so did you use kmeans? if yes, rephrase it as "the recorded AE signals were analysed using k-means". You mention [30,31]: is that a particular implementation used in those references which requires its citation in that sentence precisely? Because kmeans was proposed by other authors than [30,31]. Please clarify.

Q11. Line 208 i learn that "Waveform Time Domain, Waveform Time Domain
(Autocorrelation), fast Fourier transform (FFT Real) and Waveform Continuous Wavelet" were used to "analyse" AE signals. Does it mean you use some features extracted from AEwin software (see Q9) and then plot those time-frequency transforms to get insights about the signals content?
Maybe a flowchart or a better description of the outline of the paper can be useful to clarify the structure of the paper.

Q12. Line 255 "were preliminarily clustered using the non-hierarchical method for AE signal clustering – k-means" can be replaced by "were preliminarily clustered using kmeans".

Q13. Line 256 "the signals were divided into five classes" can you justify the number of clusters?

Q14. Can you justify the subset of features used?

Q15. Line 261 "It is easy to notice" please remove "easy"

Q16. "the great majority of AE signals from classes 1–5 occur just after the maximum force is reached and later, i.e. after the main crack has propagated"

-> To my experience the kmeans often gives clusters starting closely in time, because kmeans does not consider the fact that points are ordered. If you run kmeans on the same dataset but my changing the order you get the same partition (using the same initial guess of parameters). If you are interested in the damage sequence you need some additional constraints to select the right subset(s) of features that emphasizes the sequence. This is the topic of [REF X3, X4, X5 above].

Q17. Figure 8a shows that the partition of the data has been made using only one feature (signal strength). If not, the other features are almost useless because we can draw horizontal lines to get the clusters. This is also a problem of kmeans which is not found using alternatives [REF X3, X4, X5 above].
Figure 8b suffers from the same comment.

Q18. Figure 9 shows signals with saturations. In particular signals d and e.
It is probably due to a high preamp. Saturation clearly poses problem for the interpretation of features. Your statistics in table 6 are therefore highly impacted.

Q19. Figure 11 (time frequency representations) is not of good quality. A zoom on the useful parts of the figure should be done.

Author Response

Thank you very much for the substantive comments, which I mostly included and changed in the text of our article.
Answers to the questions below:

Ad. Q1

This statement is supported by studies of the behavior of individual constituent materials of the laminate and of the laminate as an independent material. This was also confirmed by tests performed on polymer-fiber laminates, which I had tested earlier.

Ad. Q2-4, Q9-14, Q16

Yes, you are right, the k-means method is the first approximation, it helps to identify destructive processes, especially when we have a very large amount of data that needs to be analyzed. The most important advantage of this method is the speed of data processing and analysis. The disadvantage, we must know the approximate number of destructive processes. In this case, based on the Hull model for laminates and the Griffith model, we can determine the occurring destructive processes. The research also uses other grouping methods (Forgy, NNC, FCM, GMD, GK, HMM, ARHMM) to isolate destructive subprocesses based on a full statistical-mathematical approach to divide into

K-means is a standard cluster analysis algorithm, in which the value of parameters determining the number of groups to be extracted from a data set is initially determined. Representatives are randomly selected so that it is important that they are as far apart as possible. Selected components are the seedbed of groups (prototypes). In the next step, each component of the set is assigned to the nearest group. Initial groups are designated at this stage. In the next step, a centre is calculated for each group based on the arithmetic mean of the coordinates of the objects assigned to a group. Then all objects are considered and reallocated to the nearest (in respect of the distance from individual centroids) group. New group centres are designated until the migration of objects between clusters ceases. According to the same principle, the assignment correctness of particular objects to particular groups is checked. If in the next two runs of the algorithm there is no change in the division made (then it is said that the stabilization is achieved), the processing is finished. In this method, the number of groups is constant and consistent with the k parameter, only the group object assignment can be changed.

Yes this is just a non-standard approach.

The exact approach algorithm for determining groups is given in the paper : Krampikowska, A .; Dzioba, I .; Pała, R .; Swit, G. The Use of the Acoustic Emission Method to Identify Crack Growth in 40CrMo Steel. Materials. 2019, 12 (13), 2140-2154.

Ad. Q5

Thank you for your attention. I corrected in the text.

Ad. Q6

Yes. We can treat this parameter as energy

Ad. Q7

Thank you very much for attention. I corrected in the text

Ad. Q 8-9

Thank you very much for attention. Yes, I believe that the functions extracted from this transformation can be useful for detecting defects. No, the wavelet transformation that creates the 3D map is not used for clustering. It was used to confirm and possibly use it for later assessment of the condition of the tested material.

Ad. Q15

Thank you very much. I corrected in the text

Ad. Q 17

The choice of the SS parameter was dictated only by the illustration of the occurrence of EA signals (divided into groups) when loading the sample. Therefore, the loading function was only recorded when the signal appeared. The use of another AE parameter will not change the appearance of the graph qualitatively.17.

Ad. Q18.

All AE measurements were made with the same apparatus settings and gain of AE sensor signals.

Ad. Q19.

The scale of the drawings is adapted to the maximum values of time and frequency. For comparison, the scale is the same for all drawings.

Reviewer 4 Report

This manuscript presents a broad study of the  different fracture mechanism in an Al-Ti composites prepared by explosion welding. The overall idea of the article is a novelty and will be a very interesting topic for the audience of this journal. These results add a novelty to current knowledge about Al-Ti composites. The abstract is self-explanatory and totally reflects the content of the article. The Materials and Methods section is well organized, the discussion section clearly states this discovery and therefore points out the added value of this research. Conclusions are well presented and describe the advances that were made as an outcome of the present paper, together with the perspectives for future research in the field.

Author Response

Thank you very much for your valuable comments and assessment of our article.
We introduce some additional information into paper text according to Reviewers comments. 

Round 2

Reviewer 1 Report

Dear author, it's very sad that not all of the reviewer’s comments were corrected:

"The quality of Figures (3-12) needs to be improved in order to clear the numeric and text information.

Figures 10 and 11 should be merged".

Please improve an article according this comments or write the remark why you do not acchiew this comments.

Author Response

Dear Reviewer
Inadvertently, I haven't really considered all your recommendations before.
Currently, according to your recommendations, I have combined drawings 10 and 11, creating separate drawings for individual classes.
According to your comments, I have enlarged and corrected imperfections in Figures 3-15. I made changes in the article.
Thank you very much for your valuable comments and recommendations.

Reviewer 3 Report

Authors have taken my remarks into account.

Author Response

Dear Reviewer
Thank you very much for substantive comments and recommendations

Round 3

Reviewer 1 Report

The authors have improved their manuscript by considering all comments.